# Heart function enhancement by an Nrf2-activating antioxidant in acute Y-strain Chagas disease, but not in chronic Colombian or Y-strain

**Hilton Antônio Mata-Santos[1]\*, Camila Victória Sousa Oliveira[2], Daniel F. Feijo[2], Daniel Figueiredo Vanzan[1], Glaucia Vilar-Pereira[3], Isalira P. Ramos[4], Vitor Coutinho Carneiro[5], Oscar Moreno-Loaiza[6], Jaline Coutinho Silverio[3], Joseli Lannes-Vieira[3], Emiliano Medei[6], Marcelo T. Bozza[2], Claudia N. Paiva [2]\***

1 Faculdade de Farmácia, Universidade Federal do Rio de Janeiro (UFRJ), Rio de Janeiro, Brazil, **2** Instituto de Microbiologia Paulo de Goes, UFRJ, Rio de Janeiro, Brazil, **3** Oswaldo Cruz Institute, Oswaldo Cruz Foundation, Rio de Janeiro, Brazil, **4** Centro Nacional de Biologia Estrutural e Bioimagem, UFRJ, Rio de Janeiro, Brazil, **5** Instituto de Bioquímica, UFRJ, Rio de Janeiro, Brazil, **6** Institute of Biophysics Carlos Chagas Filho, UFRJ, Rio de Janeiro, Brazil

\* hilton@pharma.ufrj.br (HAM-S); cnpaiva@micro.ufrj.br (CNP)

**Data Availability Statement:** All relevant data are within the manuscript and its Supporting information files.

## Abstract

Oxidative stress promotes *T. cruzi* growth and development of chronic Chagas heart dysfunction. However, the literature contains gaps that must be fulfilled, largely due to variations in parasite DTU sources, cell types, mouse strains, and tools to manipulate redox status. We assessed the impact of oxidative environment on parasite burden in cardiomyoblasts and the effects of the Nrf2-inducer COPP on heart function in BALB/c mice infected with either DTU-II Y or DTU-I Colombian *T. cruzi* strains. Treatment with antioxidants CoPP, apocynin, resveratrol, and tempol reduced parasite burden in cardiomyoblasts H9C2 for both DTUI- and II-strains, while $H_2O_2$ increased it. CoPP treatment improved electrical heart function when administered during acute stage of Y-strain infection, coinciding with an overall trend towards increased survival and reduced heart parasite burden. These beneficial effects surpassed those of trypanocidal benznidazole, implying that CoPP directly affects heart physiology. CoPP treatment had beneficial impact on heart systolic function when performed during acute and evaluated during chronic stage. No impact of CoPP on heart parasite burden, electrical, or mechanical function was observed during the chronic stage of Colombian-strain infection, despite previous demonstrations of improvement with other antioxidants. Treatment with CoPP also did not improve heart function of mice chronically infected with Y-strain. Our findings indicate that amastigote growth is responsive to changes in oxidative environment within heart cells regardless of the DTU source, but CoPP influence on heart parasite burden in vivo and heart function is mostly confined to the acute phase. The nature of the antioxidant employed, *T. cruzi* DTU, and the stage of disease, emerge as crucial factors to consider in heart function studies.

**Funding:** This work was supported by the Coordenação de Aperfeiçoamento de Pessoal de Nível Superior Brasil (CAPES) – UFRJ - Instituto de Microbiologia Paulo de Góes/UFRJ to CNP and MTB; by the Conselho Nacional de Desenvolvimento Científico e Tecnológico, CNPq, CNPq1A to MTB, CNPq1B to JLV and CNPq 1C to EM); by the e Fundação Carlos Chagas Filho de Amparo à Pesquisa do Rio de Janeiro FAPERJ (PROCESSO E-26/ E-26/010.02425/2019 FAPERJ/ RJ to MTB). The funders had no role in studying design, data collection and analysis, decision to publish, or preparation of the manuscript.

**Competing interests:** The authors have declared that no competing interests exist.

## Author summary

Chagas disease, caused by Trypanosoma cruzi, can lead to serious heart complications, known as Chagas cardiomyopathy. This study investigates the role of oxidative stress on the growth of T. cruzi and the effects of antioxidants, particularly protoporphyrin cobalt (CoPP), on heart health using two distinct strains of the parasite (Y and Colombian strains). We found that CoPP and other antioxidants, including apocynin, resveratrol, and tempol, reduced parasite levels in heart cells, while oxidative stress increased it. In mice infected with the Y strain of T. cruzi, CoPP treatment during the acute phase improved heart function, reduced parasite burden, and increased survival, outperforming the standard treatment with benznidazole.

However, CoPP's benefits were mostly confined to the early stages of infection. When treatment was administered during the chronic phase, it did not improve heart function or reduce parasite levels in mice infected with the Colombian strain, despite previous studies showing that other antioxidants can be beneficial in this stage. These findings suggest that the type of antioxidant and the timing of treatment are crucial in managing Chagas disease.

This work underscores the potential of early antioxidant treatments like CoPP to protect heart function and prevent long-term damage, potentially offering better treatment strategies.

## Introduction

Oxidative stress is a distinctive feature of *Trypanosoma cruzi* infection [1]. Given that macrophages generate reactive oxygen species (ROS) to counteract infections, the finding that ROS promotes the growth of *T. cruzi* in macrophages infected with the Y strain was surprising [2,3]. This observation was made by systematically manipulating the redox status with antioxidants and through transfection with HO-1 (an antioxidant enzyme) or Nrf2 (a transcription factor orchestrating antioxidant defenses) genes. Consistent with these data, a low parasite burden was observed in macrophages unable to execute an oxidative burst due to the absence of the gp91 subunit of NOX2, following infection with either the Y strain (DTU II) [2,4] or CL-Brenner clone (DTU VI) [4]. Moreover, incubation with $H_2O_2$ prior to invasion restored amastigote growth in Nrf2-deficient macrophages, suggesting that $H_2O_2$ signals parasites to directly stimulate their growth [4]. Since ROS production is associated with a number of changes in macrophage physiology, such as availability of the labile iron pool, which are important to parasite growth [2], we assume that the oxidative cellular environment provide both direct ($H_2O_2$) and indirect (nutrients) signals to amastigote growth [1].

For cell types other than macrophages, information concerning stimulation of parasitism by ROS is limited. Pre-incubation with $H_2O_2$ enhanced *T. cruzi* CL-Brenner growth in fibroblasts [5], aligning with the observed increased proliferation of epimastigotes from the Y strain in medium containing $H_2O_2$ [6]. Parasites from the Y strain overexpressing TcCyP19-HA, a molecule that increases ROS production, have increased growth in myoblasts [7]. These data are also aligned with treatment of early infection by Y strain with the Nrf2-inducer antioxidant CoPP, which reduces parasitism not only on peritoneal macrophages, but also in the hearts of C57BL/6 mice [2]. However, while the antioxidant enzyme catalase decreased the parasite burden in cardiomyocytes infected with *T. cruzi* JG (DTU II), it did not alter the burden of cardiomyocytes infected with Col1.7G2 (DTU I) [8]. In summary, in every cell type tested,

macrophages, myoblasts, fibroblasts and cardiomyocytes [2,4,5,7,8], oxidative stress correlated with increased growth of DTU II; DTU IV behaved similarly but was only tested in macrophages [4]. On the other hand, growth of DTU I was not observed in cardiomyocytes upon oxidative stress [8]. Thus, while ROS generally enhances parasitism across various cell types and *T. cruzi* strains, its effects are not universally consistent.

Antioxidant therapy has been proposed as a co-adjuvant in chronic Chagas cardiomyopathy (CCC) [9,10] since oxidative stress is a clear contributor to tissue damage and physiological heart dysfunction. In CCC, while Nrf2 expression and nuclear translocation in heart biopsies are significantly decreased [11], indicating that heart tissues are unprotected against oxidative stress, treatment with ROS-scavenger PBN improves heart function in mice [12]. In fact, chronically infected BALB/c mice (Colombian strain, DTU I) treated with the antioxidant Nrf2/ SIRT1-SIRT3-AMPK activator resveratrol have decreased incidence of arrhythmias [13] and improved ventricular function [13]. Also, although BALB/c mice treated at chronic stage with resveratrol present decreased heart parasite burden, treatment with SOD-mimetic antioxidant tempol improved arrhythmias and ventricular function without affecting parasite burden [13]. These latter data indicates that at the chronic stage antioxidant activity fails to reduce parasitism but does not depend on reducing it to improve heart function, in alignment with the current view that CCC is independent of the parasite presence [14]. In fact, treatment of C57BL/6 mice infected with Sylvio X10/4 (DTU I) at the chronic stage with SIRT-1 activator SRT1720 or treatment of C57BL/6 mice chronically infected with Brazil strain with SIRT-3-AMPK-Nfr2 activator honokiol did not affect heart parasitism but improved ventricular function [15,16], arguing for independent effects of antioxidants on parasite burden and heart function during chronic Chagas disease.

In this study, we investigated the impact of antioxidants and hydrogen peroxide on cardiomyoblast parasite burden and the effects of the Nrf2-inducer CoPP (cobalt protoporphyrin) on heart function in BALB/c mice infected with either DTU II Y or DTU I Colombian *T. cruzi* strains. The antioxidant CoPP is known to act through Nrf2 activation to activate antioxidant defenses, particularly the expression of HO-1 (heme oxygenase-1) and was previously used by us in *T. cruzi* infection by Y strain in C57BL/6 mice, reducing parasitemia and macrophage parasitism [1]. Our findings confirm that, despite its positive impact on reducing parasite burden in cardiomyoblasts, antioxidant activity alone is insufficient to decrease the parasite burden in the heart or to enhance heart function during the chronic stage. The nature of the antioxidant employed, *T. cruzi* DTU, and the stage of disease, emerge as crucial factors to consider in heart function studies.

## Results

### Antioxidants reduce amastigote burden in cardiomyoblasts infected with Colombian (DTU I) or Y strain (DTU II)

Our previous research established that in C57BL/6 mouse peritoneal macrophages and in the human THP-1 cell line, treating Y strain trypomastigote-infected cells with the Nrf2 inducer CoPP significantly reduced parasitism [2]. This reduction was similarly observed when using various antioxidants, including NAC, resveratrol, oltipraz, pterostilbene, sulforaphane, catalase-PEG, SOD-PEG, and apocynin. Conversely, increasing parasitism was noted when cells were treated with the respiratory burst inducer PMA, $H_2O_2$, the pro-oxidant paraquat, or the HO-1 inhibitor SnPP [2]. Here, we investigated the impact of general antioxidants (NOX-2 inhibitor apocynin, SOD-mimetic tempol), Nrf2 inducers (CoPP, resveratrol) and $H_2O_2$ on the parasite load in rat cardiomyoblast H9C2 cells infected with either the DTU II Y strain or

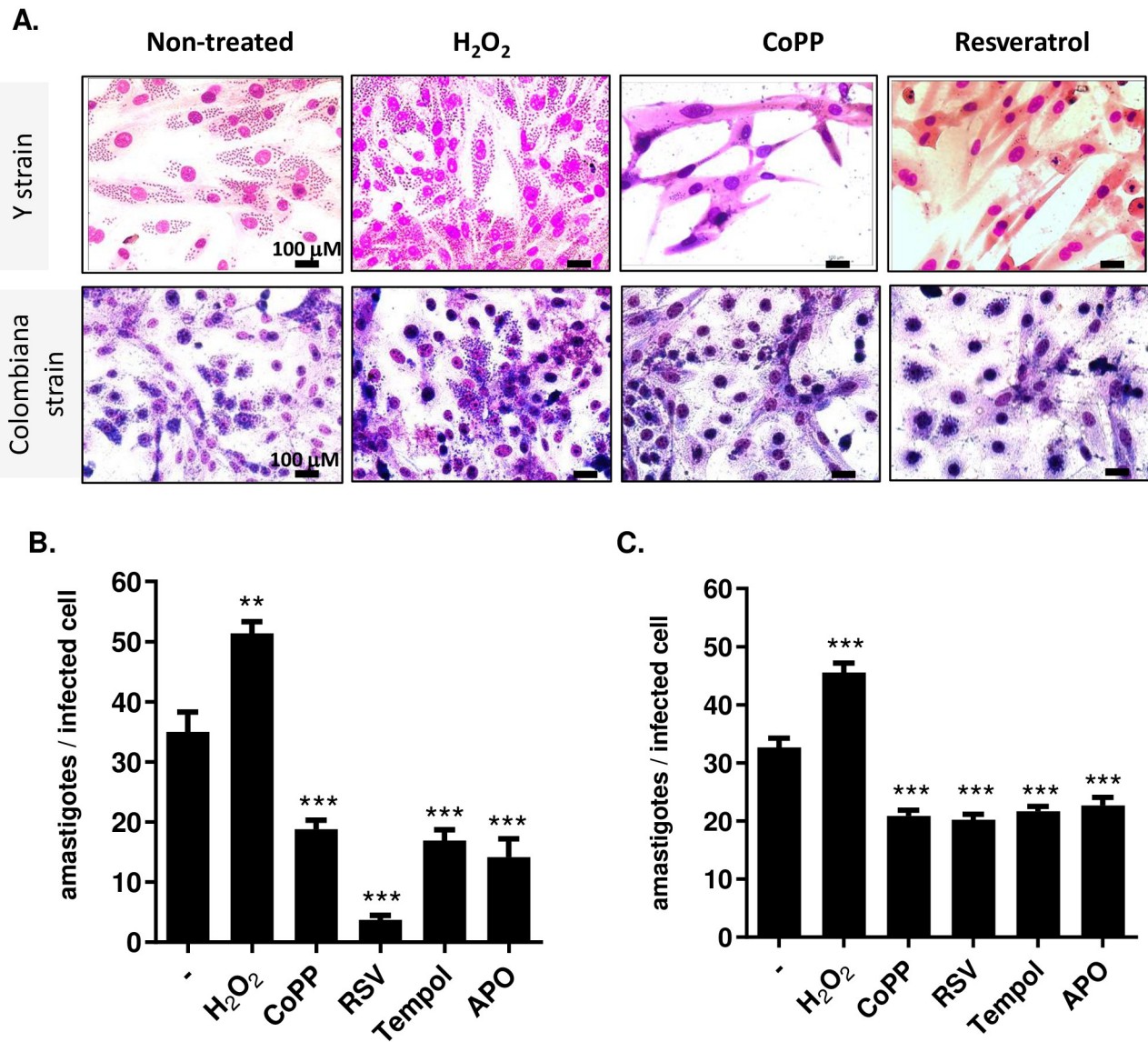

**Fig 1. Treatment with antioxidant agents reduces the parasite load in cardiomyoblast lineage cells infected with *T. cruzi* DTU II and I.** Cardiomyoblast lineage cells, H9C2, were infected at a ratio of 3 parasites per cell (3:1). (A) representative slides; cells were colored with Giemsa. (B) Y strain (DTU II) and (C) Colombian strain (DTU I) amastigote burden. After 12 hours of infection, pro- and antioxidant antioxidant agents were incubated at the following concentrations: 20μM $H_2O_2$; 50μM CoPP; 50μM resveratrol; 50μM tempol and 1mM apocynin. After 48h of treatment, the number of amastigotes per infected cell was counted. Representative experiment performed in duplicate. ** p = 0.002; ***p<0.0001.

the DTU I Colombian strain of *T. cruzi*. Treatment with $H_2O_2$ produces oxidative damage and activates redox signaling pathways [17].

Cells changed their shape and culture confluence, particularly in the presence of Y-strain infection and CoPP. We observed a significant reduction in parasitism with antioxidant treatment, whereas $H_2O_2$ treatment exacerbated the amastigote burden in H9C2 cells infected with Y and Colombian strains (Fig 1A–1C).

These findings indicate that the oxidative environment impact on parasite growth is consistent across *T. cruzi* DTUs I and II and that these effects are also applicable to cardiac cells.

## Treatment with CoPP of BALB/c mice acutely infected with Y-strain reduces heart parasitism and improves heart electrical function

The Nrf2-inducer CoPP induces HO-1 expression and reduces parasitism by acute Y strain infection in C56BL/6 mice, while HO-1 inhibitor SnPP enhances it [2], but this mouse model is not appropriate to heart studies, since infected untreated mice do not develop chronic cardiac dysfunction. Therefore, we adopted BALB/c mouse in heart studies, treating them with CoPP right after Y strain inoculation, from 0–10 days post-infection (dpi) [13].

The parasitemia remained lower than controls until 8 dpi in mice treated with CoPP, but by 11 dpi it surpassed non-treated controls (Fig 2A). Mice treated with SnPP (an inhibitor of heme oxygenase activity [18]) presented an earlier peak of parasitemia than non-treated controls. While non-treated infected mice and mice treated with SnPP died during late acute phase, mice treated with CoPP survived acute phase, despite a greater peak of parasitemia (Fig 2B).

To test whether the effects of CoPP and SnPP on the macrophage parasite burden (Y strain) of C57BL/6 mice were also present in macrophages from BALB/c mice, we examined the amastigote burden in peritoneal macrophages ex vivo. (Fig 2C). Macrophages from CoPP-treated mice had reduced amastigote burden compared to non-treated macrophages, while SnPP-treated mice presented increased macrophage parasite burden. When macrophages from BALB/c mice were infected in vitro with Y strain and treated with CoPP or SnPP, they behaved similarly to peritoneal macrophages from BALB/c mice infected with Y strain and treated in vivo with CoPP and SnPP: CoPP reduced, while SnPP increased parasitism (Fig 2D). Thus, macrophages from BALB/c mice infected with Y strain behave just like macrophages from C57BL/6 mice when treated with CoPP/ SnPP, both ex vivo and in vitro.

Hearts from CoPP-treated BALB/c mice infected with Y strain presented lower numbers of amastigotes than non-treated counterparts at 14 dpi, as well as a less intense inflammatory infiltrate (Fig 2E–2G). A lower plasma level of CK-MB, a marker of heart lesion, was found in mice treated with CoPP than in non-treated controls (Fig 2H). Liver sections from CoPP-treated BALB/c mice, on the other hand, presented increased numbers of amastigotes and similar inflammatory infiltrates compared to non-treated mice (Fig 2E, 2I and 2J). The increase in liver parasite burden, as well as non-examined tissues, might be responsible for the higher peak of parasitemia in CoPP-treated mice during acute phase. As a general outcome, these results indicate that treatment with CoPP during acute phase of infection of BALB/c mice with Y strain reduce heart parasitism and inflammation, besides allowing survival.

EKG was performed at the end of the acute phase of BALB/c mice infected with Y strain and treated with CoPP (15 dpi). Infected mice presented bradycardia, prolonged PR and QTc intervals, prolonged duration P wave (Fig 2K–2N and 2P), like we have previously shown to C57BL/6 mice, and prolonged QRS (Fig 2K and 2O). Treatment with CoPP from 0–10 dpi significantly prevented all these changes, though to a limited extension, except for prolonged QRS.

In echocardiography studies (ECHO), infected mice presented decreased stroke volume (SV) and left ventricle diameter, which did not change in response to treatment with CoPP (Fig 2Q and 2S). Thus, the effects of treatment with CoPP in heart did not affect ventricular function as much as they affected heart parasitism and EKG.

## Improvement of heart function in Y-strain acutely infected mice is mediated by both reduction of parasite burden and direct effects on heart physiology

Since we have previously found that C57BL/6 mice infected with Y strain preserve the best reduction of parasitism if we avoid extending CoPP treatment after 8 dpi, and in Fig 2 we

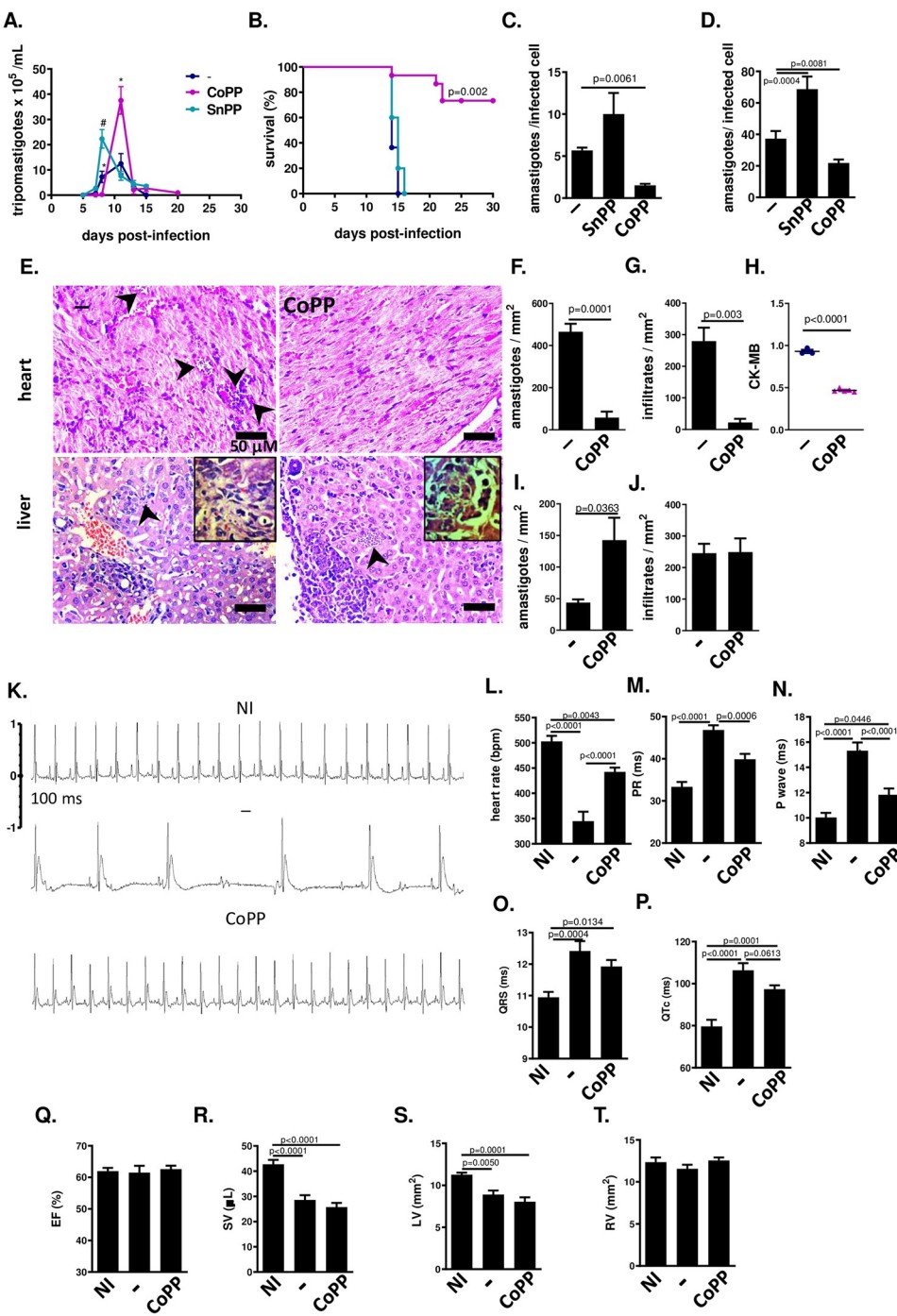

**Fig 2. Treatment with CoPP during acute Y strain infection increases survival, reduces heart parasitism and inflammation.** BALB/c mice were infected with $10^3$ blood trypomastigotes of the Y strain of *T. cruzi* and treated from days 0–10 with CoPP (5 mg/Kg, i.p.) or SnPP (5 mg/Kg, i.p.). (A) Parasitemia (n = 6–7 mice per group); (B) survival (n = 4–11 mice per group); (C) Parasite burden in peritoneal macrophages (ex vivo 15 dpi, n = 3–8 mice per group); (D) Parasite burden in peritoneal macrophages infected in vitro and treated with CoPP or SnPP (n = 20 cels per bar, from experiment of 2); (E) Heart (left ventricle) and liver histopathology (H&E slides); (F) Cardiac parasitic load (amastigote counting in H&E slides; n = 3–5 mice per group); (G) Cardiac infiltrate (leukocyte counting in H&E slides, n = 3–5 mice per group); (H) Plasma levels of the cardiac injury marker CK-MB (n = 3–5 mice per group); (I) Hepatic parasitic load (amastigote counting in H&E slides; n = 3–5 mice per group); (J) Hepatic infiltrate (leukocyte counting in H&E slides, n = 3–5 mice per group); (K) Representative EKG tracing; (L-P) EKG parameters (n = 14–18 mice per group): heart rate (RR interval); duration of PR interval; duration of P wave; duration of QRS; QT interval corrected by

heart rate; (Q-T) Echo parameters (n = 10–18 mice per group): % Ejection Fraction (stroke volume/ end diastolic volume); Stroke Volume (μL); Left Ventricle Area (mm$^2$); Right Ventricle Area (mm2), both transverse section. NI = non-infected; – = infected non-treated; CoPP = infected mice treated with CoPP; SnPP = infected treated with SnPP. Student's t test was used to find a p-value.

observed that parasitemia increased in similar conditions in CoPP-treated BALB/c mice after 8 dpi, we shortened treatment to try to avoid a late rebound of parasitemia. Also, we performed a suboptimal treatment with benznidazole (25 mg/Kg), in an order to assess the effects of partially reducing parasite burden on acute phase heart function and added CoPP to understand if adding this treatment would improve heart function.

The BALB/c mice infected with Y strain were treated from 0–8 dpi with CoPP or from 0–4 dpi with benznidazole. Parasitemia was greatly decreased in mice treated with benznidazole compared to infected non-treated mice (Fig 3A). Shortened CoPP treatment, unlike extended one, produced a peak of parasitemia similar to non-treated controls. A biomarker of heart lesion, CK-MB, was significantly decreased in infected CoPP-, benznidazole-, CoPP+ benznidazole-treated groups compared to infected non-treated mice at 15 dpi (Fig 3B).

Heart sections from BALB/c infected with Y strain and treated with CoPP presented far less amastigotes and inflammatory infiltrates than non-treated ones. Despite suboptimal, treatment with benznidazole was more effective than CoPP to reduce amastigote burden and inflammatory infiltrates and adding CoPP to benznidazole treatment did not alter efficacy (Fig 3C–3E).

The EKG from infected non-treated BALB/c mice infected with Y strain presented bradycardia, prolonged PR, QTc, QRS interval, and P-wave duration at 15 dpi (Fig 3F–3J). Treatment with CoPP (0–8 dpi) significantly prevented bradycardia and shortened PR, QRS intervals and P-wave duration more efficiently than benznidazole. Treatment with both CoPP and benznidazole presented a trend towards decreased efficiency in ameliorating heart electrical function. Together, these results indicate that the effects of CoPP to improve heart electrical function extend beyond its effects to reduce parasite burden, since despite less effective than benznidazole to reduce heart parasitism, it was more still efficient to improve heart function.

## Treatment with CoPP of BALB/c mice acutely infected with Y-strain prevents deterioration of heart function at chronic stage

Since we have found that treatment of BALB/c mice infected with Y strain from 0–8 dpi could greatly reduce heart parasitism and improve acute phase heart function, we accompanied mice these mice until the chronic stage, to test whether these alterations would be preserved. To allow their survival, these mice were infected with less trypomastigotes. During acute stage, these mice presented a delay in peak of parasitemia (Fig 4A).

At the late chronic stage (150 dpi), mice treated from 0–8 dpi presented a less prolonged QTc than untreated mice (Fig 4G), with no changes in other EKG parameters. Reduced EF (Fig 4H) and SV (Fig 4I), besides enlarged RV (Fig 4K), were observed in untreated infected mice, but not much so in CoPP treated mice. These results indicate that treatment during acute phase changes the course of infection so that prevents heart function from deteriorate.

## Treatment with CoPP of BALB/c mice chronically infected with Y-strain only slightly improved QTc

We tested whether treatment with Nrf2-inducer CoPP of BALB/c mice chronically infected with Y strain would improve its heart function, like treatment with Nrf2-inducer AMPK-

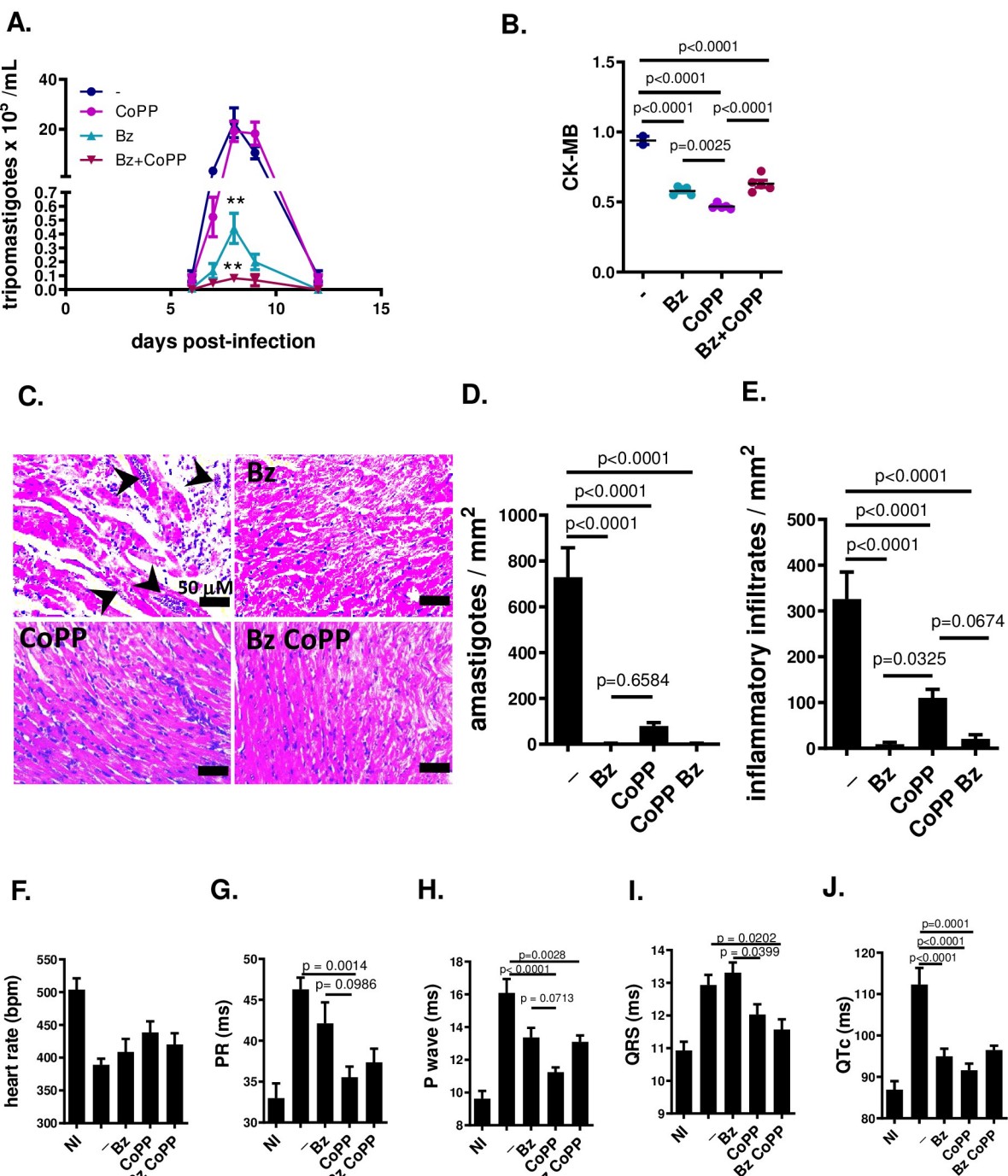

**Fig 3. Treatment with CoPP and/ or Benznidazole improves cardiac function in the acute phase of Chagas disease caused by Y strain *T. cruzi*.** BALB/c mice were infected with $10^3$ blood trypomastigotes of the Y strain of *T. cruzi* and treated from days 0–8 with CoPP (5 mg/Kg, i.p.) and/or BZ (25 mg/Kg). (A) Parasitemia (n = 5 mice per group); (B) Plasma levels of the cardiac injury marker CK-MB (n = 2–5 mice per group); (C) heart histopathology (H&E slides); (D) Cardiac parasitic load (amastigote counting in H&E slides, n = 5–7 mice per group); (E) Cardiac infiltrate (leukocyte counting in H&E slides, n = 5–7 mice per group); (F-J) EKG parameters (n = 9–10 mice per group); were performed in NI = not infected;— = infected and untreated mice; CoPP = infected mice treated with CoPP.

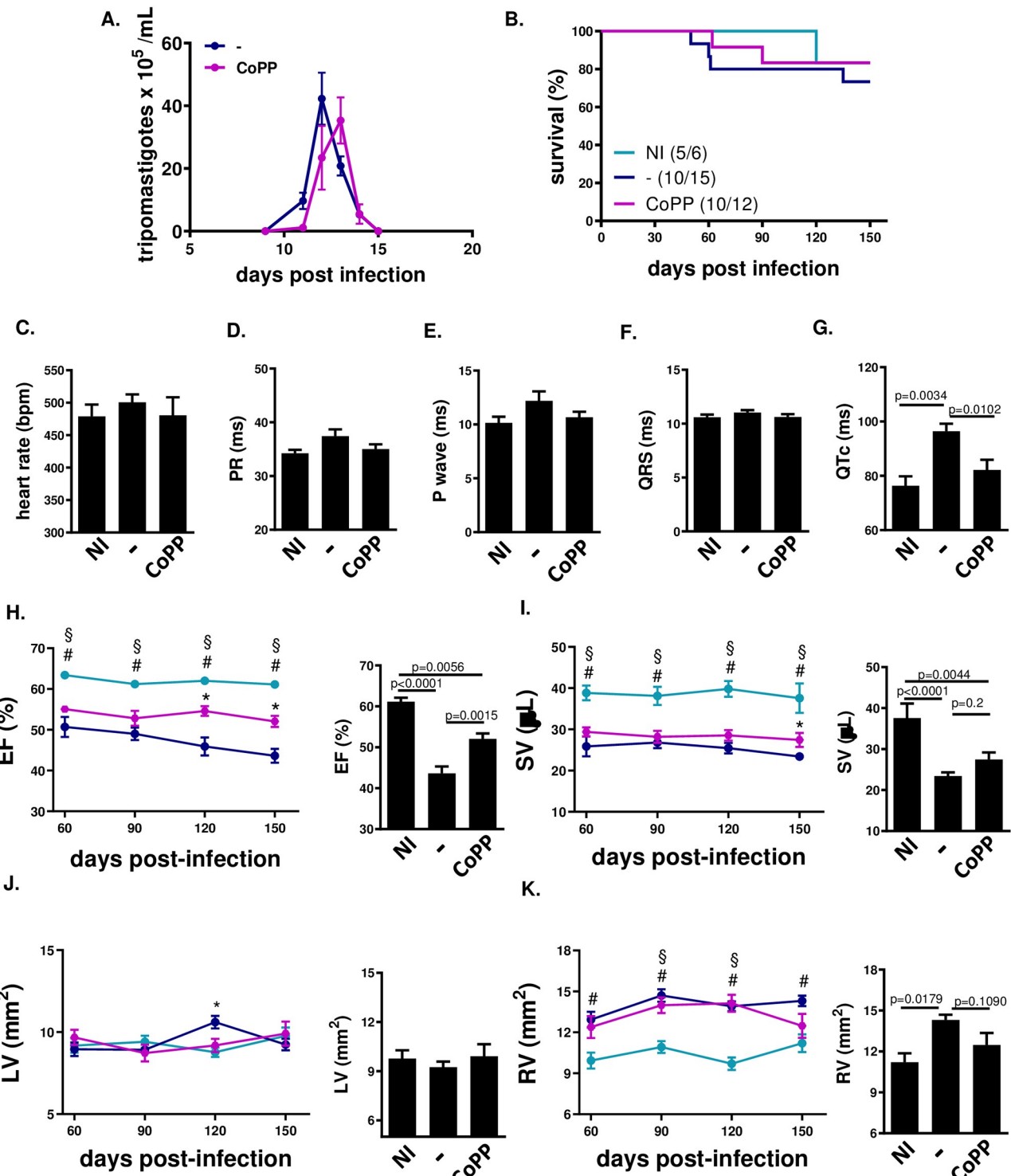

**Fig 4. Treatment with CoPP prevents QTc prolongation and improves cardiac mechanical function at later times.** BALB/c mice were infected with 50 blood trypomastigotes of the *T. cruzi* Y strain and treated daily with CoPP (5 mg/Kg, i.p.) in the period 0–8 dpi, then assessed at the chronic period of infection, from 60–150 dpi. (A) Parasitemia (n = 6–15 mice per group), (B) Survival (n = 6-15mice per group), (C-G) EKG parameters (n = 5–12 mice per group): heart rate (RR interval); duration of PR interval; duration of P wave; duration of QRS; QT interval corrected by heart rate. (H-K) Echo parameters (n-5-9 mice per group): % Ejection Fraction (stroke volume/ end diastolic volume); Stroke volume (µL); Left Ventricle area and Right Ventricle area, mm2, transverse section. NI = not infected;— = infected and untreated mice; CoPP = infected mice treated with CoPP.

activator resveratrol does [13]. Surviving infection with Y strain requires low numbers of trypomastigotes used as inoculum in BALB/c mice (50 trypomastigotes instead of $10^4$) and has been poorly studied in the literature, particularly concerning heart function. Mice were treated from 60–90 dpi with CoPP and heart function was assessed.

The chronic stage of Chagas disease after Y strain infection of BALB/c mice presented bradycardia and prolonged QTc, but not prolonged P-wave duration, QRS or PR intervals (Fig 5A–5E). Treatment with CoPP from 60–90 dpi shortened QTc but did not affect bradycardia. The ECHO of non-treated infected mice presented a decreased EF and SV, but treatment with CoPP did not consistently improve mechanical function (Fig 5F).

## Treatment of BALB/c mice chronically infected with Colombian-strain with CoPP does not reduce parasitism or improve heart function

Treatment of BALB/c mice chronically infected with Colombian strain with Nrf2-inducer and AMPK-activator resveratrol greatly improves heart function. Here, to test whether Nrf2-activation would suffice to improve heart function, we treated BALB/c mice infected with Colombian strain with CoPP from 60–90 days post-infection (dpi). Treatment with CoPP did not alter heart inflammatory infiltrates (Fig 6A and 6B) or parasite burden (Fig 6C), although the

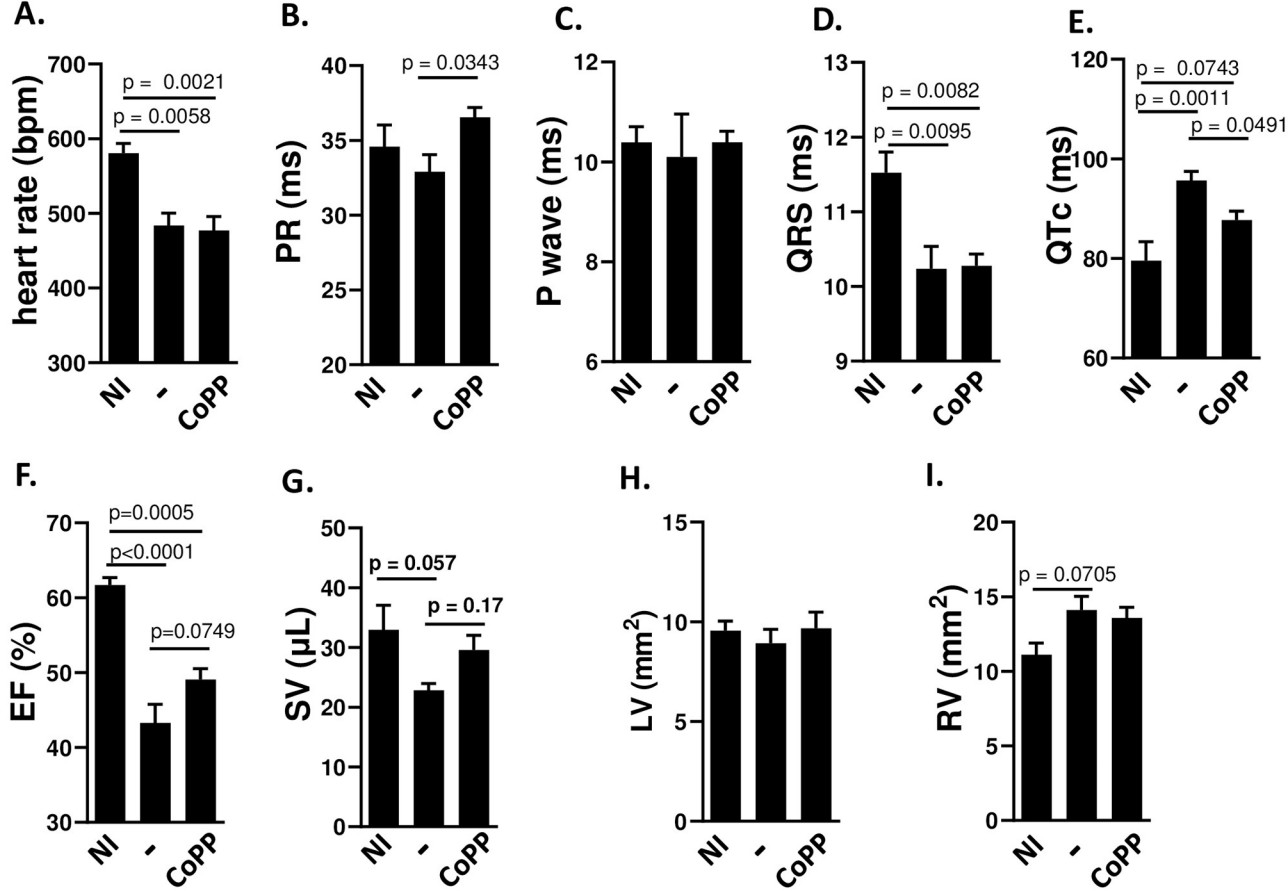

**Fig 5. Treatment with CoPP improves mechanical, but not electrical function, of mice chronically infected with Y strain.** BALB/c mice were infected with 50 blood trypomastigotes of the Y strain of *T. cruzi* and treated from days 60–90 with CoPP (5 mg/Kg, i.p.). (A-E) EKG parameters (n = 5–6 mice per group); heart rate (RR interval); duration of PR interval; duration of P wave; duration of QRS; QT interval corrected by heart rate. (F-I) Echo parameters (n-5-9 mice per group): % Ejection Fraction (stroke volume/ end diastolic volume); Stroke volume (µL); Left Ventricle area and Right Ventricle area, mm², transverse section. NI = not infected;— = infected and untreated mice; CoPP = infected mice treated with CoPP.

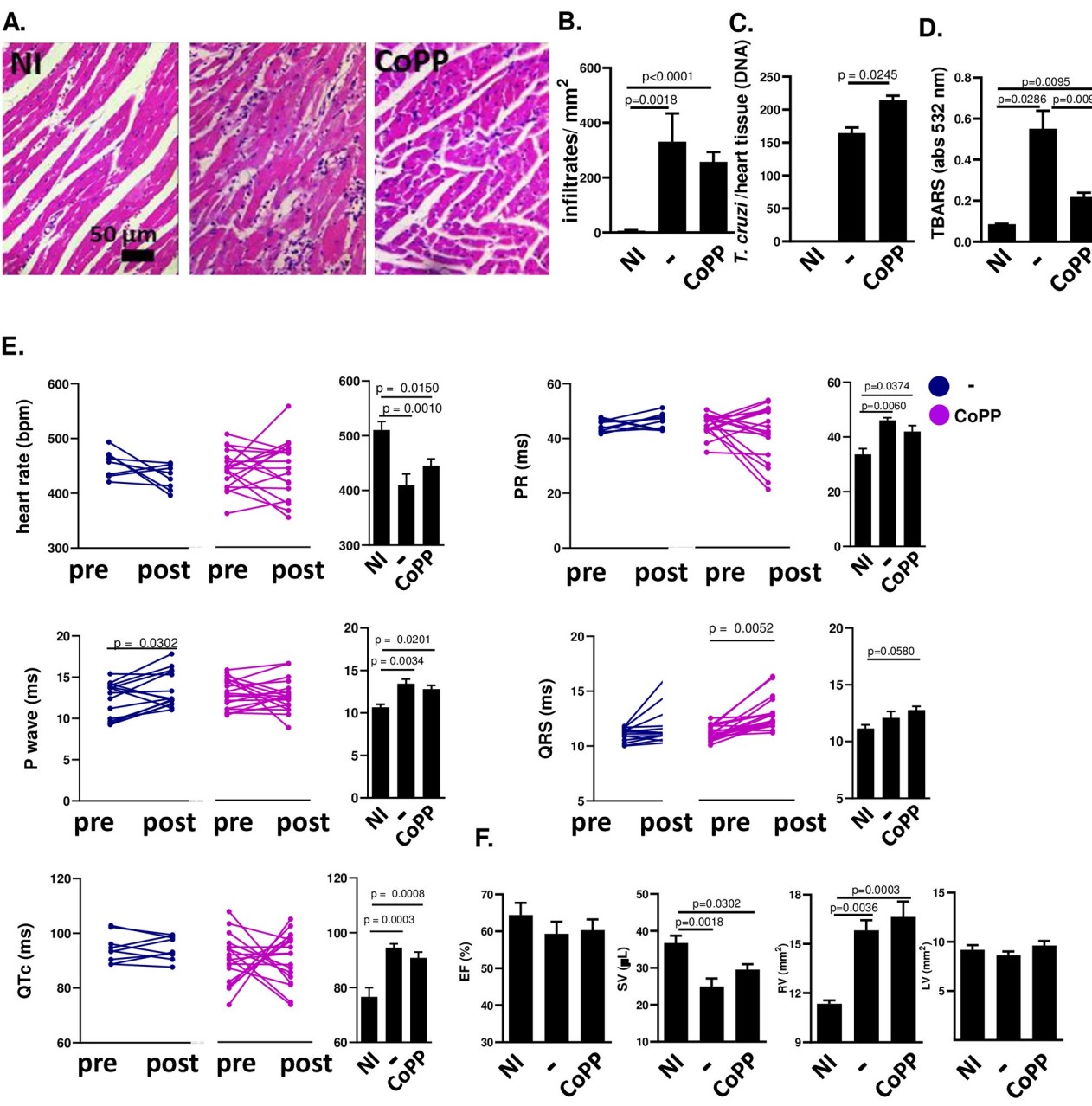

**Fig 6. Treatment with CoPP fails to improve heart function of mice chronically infected with Colombiana *T. cruzi* strain.** BALB/c mice were infected with $10^2$ blood trypomastigotes of the Colombiana strain of *T. cruzi* and treated from days 60–90 after infection with CoPP (5 mg/Kg, i.p.). (A) heart histopathology (H&E slides; n = 5–16 mice per group); (B) cardiac infiltrate (leukocyte counting in H&E slides, n = 5–16 mice per group); (C) parasite burden inferred from *T. cruzi*/ heart tissue DNA (n = 2–3 mice per group); (D) lipid peroxidation as indicative of oxidative stress, measured by TBARS (n = 4–6 mice per group); (E) Pre (60 dpi) and post (90 dpi) EKG parameters (n = 9–21 mice per group): heart rate (RR interval); duration of PR interval; duration of P wave; duration of QRS; QT interval corrected by heart rate; (F) Echo parameters (n = 4–7 mice per group): % Ejection Fraction (stroke volume/ end diastolic volume); Stroke volume (μL); Left Ventricle area and Right Ventricle area, mm$^2$, transverse section. NI = non-infected; − = infected non-treated; CoPP = infected treated with CoPP.

intensity of lipid peroxidation was reduced (Fig 6D), indicating that CoPP exerted its antioxidant effects. However, treatment with CoPP did not significantly alter electrical (Fig 6E) or ventricular heart function (Fig 6F), demonstrating that Nrf2 induction does not reverse heart dysfunction at the chronic stage.

## Discussion

The effects of oxidative stress on *T. cruzi* growth and Chagas heart disease have been the subject of recent debate [1]. Controversies have arisen due to variations in parasite strains, cell types, mouse lineages, Chagas disease stages, and the use of different drugs or genetic manipulations to alter redox status [1]. In this context, we previously reported the effects of treatment with the antioxidant CoPP on acute Y strain infection in C57BL/6 mice, which resulted in a reduction in parasite burden in both the heart and macrophages [2]. Additionally, we have observed improvements in heart function and a reduction in parasite burden in BALB/c mice chronically infected with the Colombian strain following treatment with the antioxidant resveratrol [13]. To address whether parasite burden and heart function are dependent on the type of antioxidant used and the parasite strain, we manipulated the redox status in cardiomyoblasts infected with both Y and Colombian strains. We found that both DTU II and DTU I parasites exhibited reduced growth when exposed to antioxidants of all kinds and increased growth when incubated with $H_2O_2$, indicating that both parasite types are responsive to changes in oxidative environment around cardiomyoblasts. Furthermore, we conducted experiments using the antioxidant CoPP to treat BALB/c mice infected with either the Y or Colombian strain, assessing heart function and parasite burden when relevant. Treatment with CoPP during the acute stage of Y strain infection was effective to improve heart function, but not during the chronic stage of Y strain or Colombian infection. This pattern could not be explained by the sole reduction of parasite burden.

Distinct responses to *T. cruzi* infection are evident in the mouse strains BALB/c and C57BL/6, with the latter exhibiting significantly greater resistance to both parasite growth and heart dysfunction [19,20]. Our previous research demonstrated the efficacy of CoPP treatment in reducing parasitemia and heart parasite burden in C57BL/6 mice acutely infected with the Y strain [2]. However, our current findings show that, under similar conditions, CoPP treatment extends the survival of BALB/c mice and delays the peak of parasitemia without actually reducing it. Interestingly, in BALB/c mice, we observed that CoPP treatment reduces parasite burden in the heart but increases it in the liver, while in C57BL/6 mice, it reduces parasite burden in both tissues [2]. The underlying reason for this contrasting behavior in the liver remains unexplained but may contribute to the absence of a reduction in the peak of parasitemia in BALB/c mice. It has been observed that HO-1 inducers such as CoPP behave differently in C57BL/6 and BALB/c mice during malaria infections, with BALB/c mice showing a more pronounced increase in HO-1 expression [21]. We speculate that HO-1 interference with the adaptive immune response might account for this less favorable outcome in BALB/c mice compared to C57BL/6 mice. In fact, treatment of mice with CoPP inhibits the growth of the Y strain in macrophages taken from both BALB/c and C57BL/6 mice, indicating that the higher parasitemias/ liver parasitism in CoPP-treated BALB/c mice are not due to decreased macrophage response.

We have observed a more favorable response of electrical heart function to CoPP treatment during the acute stage compared to the chronic stage of Y strain infection. During the acute stage of infection by the Y strain, the EKG pattern was significantly preserved with treatment, showing a heart rate close to normal and less pronounced increases in PR interval, P-wave duration, and QTc. However, when treatment was initiated during the chronic stage (60–90 dpi, Fig 4), it only improved the QTc interval. This lack of responsiveness to CoPP treatment aligns with observations in the chronic stage of Colombian strain infection, and it casts doubt on the potential for Nrf2 activators to effectively treat arrhythmias at this stage, despite their established reputation as cardiac tissue protectors in conditions such as diabetes and hypoxia.

Our experiments have clearly demonstrated that when treatment is administered during the acute phase of Y strain infection, both electrical and mechanical heart function are more preserved as the disease progresses to the chronic stage. This observation reinforces the idea that the severity of the acute phase plays a critical role in determining the extent of chronic heart disease, most likely because of parasite burden and inflammatory response [22]. During acute phase, treatment was very effective to reduce heart parasite burden, though a little less capable than benznidazole, and still, greatly reduced inflammatory infiltrates. A similar prevention in heart inflammation has been previously shown in an attempt to manipulate HO-1 expression by treatment with heme in mice infected with Y strain [23]. In this case however, the iron load produced a confounding factor, which most likely accounts for the increased heart parasite burden observed with heme treatment. We recall that mice acutely infected with Y strain also reacted with reduction of heart parasite burden to treatment with vitamin C [24], melatonin [25], and resveratrol [2], indicating that during acute phase, antioxidants in general act favorably in DTU II infections.

It is well established that oxidative stress is associated with increased *T. cruzi* growth of both Y strain and CL-Brenner clone in macrophages, myoblasts, and fibroblasts [1]. However, while treatment with catalase reduced parasite burden in mouse cardiomyocytes infected with the JG strain (DTU II, similar to the Y strain), it failed to do so in infections caused by the Col1.7G2 strain (DTU I). In this study, we investigated whether the oxidative environment would affect the growth of both the Colombian (DTU I) and Y strain (DTU II) in the H9C2 rat cardiomyoblast lineage. Our findings indicated that several antioxidants were capable of reducing parasite burden regardless of the strain, while $H_2O_2$ enhanced the proliferation of both strains. We believe that the genetic diversity within the Colombian stock, compared to the single clone Col1.7G2 obtained from this stock, may account for the conflicting results observed in our study and that of others, underscoring the importance of extending these investigations to as many *T. cruzi* strains as possible. The frequency of non-redox responsive trypomastigotes is unknown, and there is even a report of increased susceptibility of NOX2--deficient peritoneal macrophages to *T. cruzi* Dm28c, although this finding is hard to be reconciled with others.

The failure to improve heart function in BALB/c mice chronically infected with the Colombian or Y strain and treated with CoPP stands in stark contrast to the reversal of heart dysfunction achieved with resveratrol in the same model [13]. The reduction in lipid peroxidation observed with CoPP treatment indicates that it indeed exhibited its expected antioxidant effects. These results highlight that merely having antioxidant properties is insufficient to improve heart function during the chronic phase of Colombian infection. One might argue that resveratrol's ability to improve heart function stems from its reduction of heart parasite burden, whereas CoPP did not affect it. However, we previously demonstrated that tempol, a SOD-mimetic antioxidant, improved heart function without altering heart parasite burden [13]. The reasons for CoPP's failure to improve heart function, as resveratrol and tempol did, remain unknown. It is possible that the difference lies in the source of ROS targeted by different antioxidant treatments. These results show that the kind of antioxidant chose to treat infected mice is fundamental to its success to improve cardiac function.

We identified several limitations in our study. Although our aim was to test DTU I and II, we only used a representative strain from each. Only when the results are consistent between the two strains, it suggests a general trend across different *T. cruzi* strains, particularly regarding parasite burden in cardiomyoblasts treated with antioxidants. Furthermore, since we did not treat BALB/c chronically infected with the Y strain using resveratrol, we cannot determine whether this infection can be ameliorated by other, more suitable antioxidants. In this study, we sought to unravel variables such as *T. cruzi* source, stage of infection, and infected cell type

that might contribute to the paradoxical findings reported in the literature. A more extensive, systematic investigation would be necessary to predict whether infection with a particular strain at a specific stage would benefit from treatment in terms of parasite burden or heart function, with the aim of developing broader, more universally applicable treatment strategies.

Several lines of evidence point to the benefits of using antioxidants in CCC: the oxidative damage to heart tissue caused by infection; the weakened antioxidant defenses; the improvement produced by antioxidants in electrical and mechanical heart function. The risk of rebound infections do not appear to be high, as distinct treatments with antioxidants attempted were unable to undermine the immunity that keeps infection at hold. Nevertheless, the distinct kinds of antioxidants do not work the same, neither do they act over distinct sources of parasites, or at distinct stages of disease equally. Here we show that antioxidants act to reduce cardiomyoblast parasite burden regardless of the DTU source, but one of them, Nrf2-activator CoPP, does not act to substantially improve heart function during chronic phase of DTU I or II. Though these results reinforce the safety of the use of antioxidants in CCC, they point to the need to determine which kind of antioxidant can be used in each DTU infection.

## Materials and methods

### Ethics statement

This study adhered to the guidelines provided by the "Guide for the Care and Use of Laboratory Animals" from the Brazilian National Council of Animal Experimentation (http://www.cobea.org.br/) and complied with Federal Law 11.794 dated October 8, 2008. All study procedures received approval from the institutional Committee for Animal Ethics of UFRJ (Licenses: IMPPG029 & IMPPG032) and Fiocruz (Licenses: 004/09 & LW10-14).

### Mice handling

BALB/c mice, both female and male aged 5–7 weeks, were sourced from the animal facilities of the Oswaldo Cruz Foundation (Fiocruz, Rio de Janeiro, Brazil) and the Universidade de São Paulo. They were housed in sterile conditions, with a stable environment of about $22 \pm 2°C$ temperature and $55 \pm 10\%$ humidity. The mice had unrestricted access to food and water and were individually marked using ear tags.

### Infection and treatment protocol

Mice were intraperitoneally infected with $10^2$ blood trypomastigote forms of the type I Colombian strain of *T. cruzi* or $50–10^2$ blood trypomastigotes of the Y strain in chronic phase studies, and $10^3$ in acute phase studies. Starting from the establishment of CCC (60 dpi), treatments were administered daily for 30 days. The treatments included oral doses of benznidazole (Rochagan, 25 mg/Kg) and i.p. CoPP or SnPP (5 mg/ Kg, Frontier Scientific) daily, which were dissolved in 0.2 M NaOH, neutralized to pH 7.2 with 1 M HCl, and adjusted to the desired concentrations with PBS.

### Histopathological analysis

Tissues from the hearts (left ventricle) and livers of infected mice were harvested and stored in formalin. These samples were then embedded in paraffin, sectioned to a thickness of 5 μm, and affixed to slides. Subsequent staining was done using the standard H&E method. Using an Olympus C21 microscope, the sections were inspected for the presence of parasite nests and documented with photographs. Parasitic infection was quantified in two ways: by calculating

the nests per 100 mm$^2$ at 8 days post-infection (dpi) when parasites were primarily found in small nests, and by counting the total number of parasites in each field at 15 dpi, using a ×400 magnification. For the latter, parasites were tallied from printed images, with at least 20 pictures evaluated per individual mouse slide. We also counted the infiltrating cells from printed images.

## TBARS

To evaluate lipid peroxidation, we measured the malondialdehyde byproduct by treating the samples with thiobarbituric acid (Sigma). We used spectrophotometry to determine Thiobarbituric acid reactive substances (TBARS) in both serum samples and cardiac tissue. In brief, serum and tissue homogenates, prepared with 0.9% NaCl, were combined with 8.1% sodium dodecylsulfate (SDS), a pH 3.5 acetic acid solution, and 0.8% thiobarbituric acid. This blend was heated for 50 minutes at 95°C. Subsequently, n-butanol was added, and the solution was centrifuged at 4000 rpm for 10 minutes. The resulting absorbance was measured at a wavelength of 532 nm.

## CK-MB

The CK-MB isoenzyme, a marker for myocardial injury, was assessed using commercial kits from Labtest (Lagoa Santa, MG, Brazil), following the methodology outlined by de Souza [26]. Serum samples were incubated with a substrate, resulting in an increase in NADPH concentration. This increase is directly related to the enzyme activity present in the samples. The analysis was performed on a Microplate Reader Benchmark spectrophotometer by Bio-Rad (Memphis, TN, USA), optimized for analyzing small amounts of mouse serum as per the manufacturer's guidelines. Measurements of optical density at 340 nm were taken at 2-minute intervals over a span of 15 minutes.

## PCR to *T. cruzi* DNA

Quantitative Polymerase Chain Reaction (qPCR) is a precise method used for detecting parasites in heart tissue samples. To begin, heart tissue samples, each weighing 10 mg, were processed for DNA extraction as per the guidelines provided by the DNeasy Blood & Tissue Kit from Qiagen. Following extraction, the genomic DNA samples underwent visualization through agarose gel electrophoresis. Subsequently, these samples were analyzed using real-time PCR, employing specific oligonucleotides: mmGAPDH-F (5' AACTTTGGCATTGTG-GAAGG 3'), mmGAPDH-R (5' ACACATTGGGGGGTAGGAACA 3'), TCZ-F (5' GCTCTTGCCCACAAGGGTGC 3'), and TCZ-R (5' CCAAGCAGCGGATAGTTCAGG 3'). The amplification process utilized Power SYBR Green 1X (Applied Biosystems) and 200 nM of each oligonucleotide within an ABI PRISM 7500 Sequence Detection System thermocycler (Applied Biosystems). This procedure involved an initial denaturation at 95°C for 10 minutes, followed by 40 cycles of denaturation at 95°C for 15 seconds and annealing at 60°C for 1 minute. A melting curve analysis was conducted post-amplification to ensure specificity by identifying non-specific amplification products. Oligonucleotide efficiency was evaluated through serial dilutions, demonstrating an efficiency range of 97% to 98%. The relative quantification of target genes was achieved by employing the comparative Ct method.

## Electrocardiography (ECG)

Mice were sedated using diazepam (10 mg/kg) and subcutaneous transducers were positioned (DII derivation). Recordings lasted 2 minutes on either Power Lab 2/20 or Power Lab 4/35

Systems connected to a PanLab Instruments bio-amplifier. The filters were set between 0.1-100Hz, and analysis was done using Scope for Windows (V3.6.10 by PanLab Instruments). More details can be found in the Online Methods section. Proper analysis of the P wave duration and conduction disorders necessitated a significant sample size for accurate statistics.

## Transthoracic echocardiography (Echo)

Under isoflurane anesthesia (2% in oxygen), mice were prepared in the precordial area with depilatory cream. Examinations utilized a 30 Mhz transducer connected to a Vevo 770 Ultrasound device (by Visual Sonics, Canada). The Left Ventricle Ejection Fraction (LVEF) was deduced using Simpson's method due to its suitability for CD heart geometry and its frequent application in CD patient assessments. Measurements of both left and right ventricular areas during diastole and systole were taken in B mode, focusing on the papillary muscles' level.

## Statistical analyses

Comparisons relied on unpaired Mann-Whitney U-tests (for two groups). For multiple comparisons, one-way ANOVA with Tukey correction was used. For pre-versus-post analyses, we performed Wilkoxon Signed Rank test. To determine differences in arrhythmia incidence among groups, Fisher's exact t-test was employed. A p-value below 0.05 was considered significant, and such values are highlighted next to the respective groups in figures.

## Supporting information

**S1 Table. Descriptive data of the values used to build graph (Fig 1).**
(XLSX)

**S2 Table. Descriptive data of the values used to build graph (Fig 2).**
(XLSX)

**S3 Table. Descriptive data of the values used to build graph (Fig 3).**
(XLSX)

**S4 Table. Descriptive data of the values used to build graph (Fig 4).**
(XLSX)

**S5 Table. Descriptive data of the values used to build graph (Fig 5).**
(XLSX)

**S6 Table. Descriptive data of the values used to build graph (Fig 6).**
(XLSX)

## Author Contributions

**Conceptualization:** Hilton Antônio Mata-Santos, Camila Victória Sousa Oliveira, Joseli Lannes-Vieira, Emiliano Medei, Marcelo T. Bozza, Claudia N. Paiva.

**Data curation:** Hilton Antônio Mata-Santos, Camila Victória Sousa Oliveira, Daniel F. Feijo, Daniel Figueiredo Vanzan, Glaucia Vilar-Pereira, Isalira P. Ramos, Vitor Coutinho Carneiro, Oscar Moreno-Loaiza, Jaline Coutinho Silverio, Claudia N. Paiva.

**Formal analysis:** Hilton Antônio Mata-Santos, Camila Victória Sousa Oliveira, Daniel F. Feijo, Daniel Figueiredo Vanzan, Glaucia Vilar-Pereira, Isalira P. Ramos, Vitor Coutinho

                    Nrf2-activator effects on Chagas disease cardiomyopathy

Carneiro, Oscar Moreno-Loaiza, Jaline Coutinho Silverio, Emiliano Medei, Marcelo T. Bozza, Claudia N. Paiva.

**Funding acquisition:** Joseli Lannes-Vieira, Emiliano Medei, Marcelo T. Bozza, Claudia N. Paiva.

**Investigation:** Hilton Antônio Mata-Santos, Camila Victória Sousa Oliveira, Daniel F. Feijo, Daniel Figueiredo Vanzan, Glaucia Vilar-Pereira, Isalira P. Ramos, Vitor Coutinho Carneiro, Oscar Moreno-Loaiza, Jaline Coutinho Silverio, Joseli Lannes-Vieira, Emiliano Medei, Marcelo T. Bozza, Claudia N. Paiva.

**Methodology:** Hilton Antônio Mata-Santos, Camila Victória Sousa Oliveira, Daniel F. Feijo, Daniel Figueiredo Vanzan, Glaucia Vilar-Pereira, Isalira P. Ramos, Vitor Coutinho Carneiro, Jaline Coutinho Silverio, Joseli Lannes-Vieira, Emiliano Medei, Marcelo T. Bozza, Claudia N. Paiva.

**Project administration:** Claudia N. Paiva.

**Resources:** Joseli Lannes-Vieira, Emiliano Medei, Marcelo T. Bozza, Claudia N. Paiva.

**Software:** Hilton Antônio Mata-Santos, Camila Victória Sousa Oliveira, Claudia N. Paiva.

**Supervision:** Joseli Lannes-Vieira, Emiliano Medei, Marcelo T. Bozza, Claudia N. Paiva.

**Validation:** Hilton Antônio Mata-Santos, Camila Victória Sousa Oliveira, Daniel F. Feijo, Daniel Figueiredo Vanzan, Glaucia Vilar-Pereira, Isalira P. Ramos, Vitor Coutinho Carneiro, Oscar Moreno-Loaiza, Jaline Coutinho Silverio, Joseli Lannes-Vieira, Emiliano Medei, Marcelo T. Bozza, Claudia N. Paiva.

**Visualization:** Hilton Antônio Mata-Santos, Camila Victória Sousa Oliveira, Daniel F. Feijo, Daniel Figueiredo Vanzan, Glaucia Vilar-Pereira, Isalira P. Ramos, Vitor Coutinho Carneiro, Oscar Moreno-Loaiza, Jaline Coutinho Silverio, Joseli Lannes-Vieira, Emiliano Medei, Marcelo T. Bozza, Claudia N. Paiva.

**Writing – original draft:** Hilton Antônio Mata-Santos, Claudia N. Paiva.

**Writing – review & editing:** Hilton Antônio Mata-Santos, Claudia N. Paiva.

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
