## [Decision Letter · Decision Letter 0]

25 Jun 2024

Dear Dr Paiva,

Thank you very much for submitting your manuscript "Heart Function Enhancement with Nrf2-Activating Antioxidant: Benefits in Acute Y-Strain Chagas Disease, Not in Chronic Colombian Strain" for consideration at PLOS Neglected Tropical Diseases. As with all papers reviewed by the journal, your manuscript was reviewed by members of the editorial board and by several independent reviewers. In light of the reviews (below this email), we would like to invite the resubmission of a significantly-revised version that takes into account the reviewers' comments. 

Please implement all requirements and recommendations of Reviewers 1 to 5 and consider several reviewer comments the writing throughout the manuscript needs significant attention, in addition to line numbers. The authors are required to use a professional writing service because two reviewers have criticised the quality and clarity of writing.

We cannot make any decision about publication until we have seen the revised manuscript and your response to the reviewers' comments. Your revised manuscript is also likely to be sent to reviewers for further evaluation.

Sincerely,

Michael W Gaunt, PhD

Academic Editor

Claudia Brodskyn

Section Editor

# Reviewer 5

The statistics are below professional standard. The T-test risks Type 1 error and multivariate tests are required. More generally unless parametric statistics conform to the normal distribution they cannot be used. The authors should seek to implement either the Mann-Whitney U, Wilcoxon Signed Rank or Kruskal-Wallace tests rather than large numbers of T-tests, where appropriate, and need statistical support.

Reviewer's Responses to Questions

**Key Review Criteria Required for Acceptance?**

**Methods**

-Are the objectives of the study clearly articulated with a clear testable hypothesis stated?

-Is the study design appropriate to address the stated objectives?

-Is the population clearly described and appropriate for the hypothesis being tested?

-Is the sample size sufficient to ensure adequate power to address the hypothesis being tested?

-Were correct statistical analysis used to support conclusions?

-Are there concerns about ethical or regulatory requirements being met?

Reviewer #1: To be more precise, add information on how many times a day the treatment doses were given. For example, was the dose of 5mg/kg administered once a day, or were there two administrations per day until reaching the dose of 5mg/kg?

In the statistical analyses, why was the Student's t-test used for multiple comparisons? In analyses involving more than 2 groups, the use of the Student's t-test increases the risk of Type I error, as multiple t-tests do not undergo statistical correction. The more appropriate approach would be to use ANOVA, followed by post-hoc tests for multiple comparisons with correction after establishing significance. Additionally, it was not explicitly stated in the methodology, but was a normality test of the data conducted before applying a parametric test? If so, please include this information

Reviewer #2: This is an experimental study that sought to assess the impact of redox status on parasite burden in cardiomyoblasts and the effects of the Nrf2-inducer COPP on heart function in BALB/c mice infected with either DTU-II Y or DTU-I Colombian T. cruzi strains. Treatment with antioxidants CoPP, apocynin, resveratrol, and tempol reduced parasite burden in cardiomyoblasts for both DTUI- and II-strains, while H2O2 increased it. CoPP treatment improved electrical heart function when administered during acute stage of Y-strain infection, coinciding with an overall trend towards increased survival and reduced heart parasite burden. These beneficial effects surpassed those of trypanocidal benznidazole, implying that CoPP directly affects heart physiology. CoPP treatment had beneficial impact on heart systolic function when started during chronic infection with Y-strain, an effect also achieved when performed during acute and evaluated during chronic stage. No impact of CoPP on heart parasite burden, electrical, or mechanical function was observed during the chronic stage of Colombian-strain infection, despite previous demonstrations of improvement with other antioxidants. Our findings indicate that amastigote growth is responsive to change in redox status within heart cells regardless of the DTU source, but CoPP influence on heart parasite burden in vivo and heart function is mostly confined to the acute phase. 

The authors concluded that the nature of the antioxidant employed, T. cruzi DTU, and the stage of disease, emerge as crucial factors to consider in heart function studies.

This is an interesting study. I had some diifficulty following the text that copuld be improved and simplified.

Reviewer #3: The methods used were sufficient to achieve the objectives.

Reviewer #4: The study objectives are clearly articulated, with an elegant, well-defined, testable hypothesis. All ethical and regulatory requirements were thoroughly met, with no concerns noted.

**Results**

-Does the analysis presented match the analysis plan?

-Are the results clearly and completely presented?

-Are the figures (Tables, Images) of sufficient quality for clarity?

Reviewer #1: The analysis matches the plan.

Results are in general clearly presented.

Figures are of good quality

Reviewer #2: (No Response)

Reviewer #3: The results achieved the initial proposed objective. However, the addition of new data would be interesting for the robustness of the manuscript.

Reviewer #4: 1. Regarding Figure 1, there is an important difference in the confluence of cells between the figures, particularly in CoPP. How did the authors explain this?

2. In Figure 2E, which section or region of the organs did the authors use? This should be clarified.

3. In Figure 3, the authors administered a suboptimal treatment with benznidazole (25 mg/kg) and added CoPP to determine if this combination would improve heart function. Did they perform the echocardiography analysis?

4. In Figure 4, the author should provide a more detailed explanation of the figure, letter by letter.

5. In Figure 6, the authors stated that treatment with CoPP did not alter heart fibrosis, but they only assessed collagen deposition. Could the authors analyze fibronectin or other extracellular matrix proteins?

**Conclusions**

-Are the conclusions supported by the data presented?

-Are the limitations of analysis clearly described?

-Do the authors discuss how these data can be helpful to advance our understanding of the topic under study?

-Is public health relevance addressed?

Reviewer #1: Conclusions are supported by the data

Limitations are generally discussed, but the inclusion of a Limitations section on the discussion would improve the manuscript.

Data are discussed in the context of relevant literature.

Public health relevance is addressed. It would be important that authors discuss possile translational implications of their experimental findings, if any

Reviewer #2: (No Response)

Reviewer #3: The conclusion of the manuscript is in accordance with the data presented.

Reviewer #4: The conclusions drawn in the study are well-supported by the comprehensive data presented. 

The authors should thoroughly explore the limitations of the study.

**Editorial and Data Presentation Modifications?**

Reviewer #1: In the introduction, could you briefly include more information about CoPP, what it is, and whether it is used in other situations, etc.

In Figure 6, wasn't it expected that the infection with the Colombian strain would cause a reduction in EF? please discuss this

In Figure 6, only the measurement of the RV was performed. Why wasn't the measurement of the LV done, as it was with the Y strain?

Reviewer #2: (No Response)

Reviewer #3: The authors must add the bar scale in the figure 1 A and the scale value in the figure 2E.

Reviewer #4: Sometimes the authors use rats, other times mice. 

In echocardiography studies, the authors did not find a significant change in ejection fraction with the treatment. The authors should further explore and explain this result.

**Summary and General Comments**

Reviewer #1: This is an original work that tests the hypothesis that the use of an Nfr2-inducer (CoPP) during acute or chronic infection could have beneficial effects on mouse infected with different strains of T. cruzi. The research is based on literature data suggesting that oxidative stress is important for the growth of the parasite. Among the findings, the beneficial effects of using CoPP seem to be present in the acute phase, without showing significant differences in the chronic phase of the disease. The work is well-constructed, supported by literature data, and includes previous work from the group.

Reviewer #2: (No Response)

Reviewer #3: It is an exciting manuscript showing the antioxidants benefits in acute Y Strain infection, focusing on heart function enhancement. The article presents an interesting idea concerning the Nrf2-HO1 axis in Y-strain infection. However, some points can be better discussed. 

Major points:

1) Although Shan and colleagues (2006) as well as other articles have shown that the effect of CoPP on HO-1 induction can be dependent on Nrf2, there is a robust literature showing that CoPP can be an inducer of HO-1. To elucidate this issue, it would be interesting for the authors to show the activation of Nrf2 (e.g. luciferase assay), and Nrf2 and HO1 expression (e.g. western blotting). The presentation of the Nrf2-HO-1 axis would make the manuscript more robust.

2) A comparison with the classical Nrf2 activator (DMF or MMF) and the classical HO-1 activator (heme, in the appropriate concentration) would be interesting.

3) The authors stated in the manuscript that they changed the redox status when they treated the cells with H2O2. However, the statement is very weak. To evaluate the redox status, authors should evaluate the production of reactive oxygen species (e.g. DCF, CellRox...) and/or antioxidant activity and/or oxidative damage (shown only in figure 6). I encourage the authors to evaluate the redox balance, as it would be interesting to understand how the non-damaging oxidative burst occurs, and which reactive oxygen species are involved. 

2) The use of H2O2 is very superficial, because the production of reactive oxygen species can occur by different sources and in different compartments, therefore it would be interesting to use selective NOX inducers.

Minor points:

1) It would be interesting to discuss that SnPP also inhibits HO-2. ZnPP is more selective for HO-1.

2) It was not clear why the authors only treated the cells with Colombian strain chronically. Although there were promising results in figure 1, it was only in the last figure that the authors returned to presenting the data involving the Colombian strain.

3) The authors should review some typos.

4) The authors must add the bar scale in the figure 1A and the scale value in the figure 2E.

Reviewer #4: This study represents an important effort in exploring the influence of redox status on parasite burden in cardiomyoblasts, as well as evaluating the impact of the Nrf2-inducer COPP on heart function in BALB/c mice infected with either DTU-II Y or DTU-I Colombian T. cruzi strains. The novelty of these findings contributes significantly to the current literature on Nrf2-activating antioxidants. The study's importance is highlighted by its potential implications for advancing our understanding of Chaga disease physiopathology.

PLOS authors have the option to publish the peer review history of their article (what does this mean?). If published, this will include your full peer review and any attached files.

Reviewer #1: No

Reviewer #2: No

Reviewer #3: Yes: João Alfredo de Moraes

Reviewer #4: No
---

## [Decision Letter · Decision Letter 1]

8 Oct 2024

Dear Dr Paiva,

We are pleased to inform you that your manuscript 'Heart Function Enhancement with Nrf2-Activating Antioxidant in Acute Y-Strain Chagas Disease, Not in Chronic Colombian or Y-Strain' has been provisionally accepted for publication in PLOS Neglected Tropical Diseases.

The manuscript is a very interesting study contrasting DTU-I and DTU-II and all revisions are in place.

Best regards,

Michael W Gaunt, PhD

Academic Editor

Claudia Brodskyn

Section Editor

<style type="text/css">p.p1 {margin: 0.0px 0.0px 0.0px 0.0px; line-height: 16.0px; font: 14.0px Arial; color: #323333; -webkit-text-stroke: #323333}span.s1 {font-kerning: none

</style>

Reviewer's Responses to Questions

**Key Review Criteria Required for Acceptance?**

**Methods**

-Are the objectives of the study clearly articulated with a clear testable hypothesis stated?

-Is the study design appropriate to address the stated objectives?

-Is the population clearly described and appropriate for the hypothesis being tested?

-Is the sample size sufficient to ensure adequate power to address the hypothesis being tested?

-Were correct statistical analysis used to support conclusions?

-Are there concerns about ethical or regulatory requirements being met?

Reviewer #2: The authors have thoroughly revised the comments of the 5 reviewers and significantly improved the manuscript. Objectives are clearly stated and addressed. This experimental study was appropriately designed and the hypothesis tested was clearly established and addressed. The authors also reviewed the statistical comments and improved the manuscript.

Reviewer #3: (No Response)

**Results**

-Does the analysis presented match the analysis plan?

-Are the results clearly and completely presented?

-Are the figures (Tables, Images) of sufficient quality for clarity?

Reviewer #2: Results are clearly presented Figures are outstanding

Reviewer #3: (No Response)

**Conclusions**

-Are the conclusions supported by the data presented?

-Are the limitations of analysis clearly described?

-Do the authors discuss how these data can be helpful to advance our understanding of the topic under study?

-Is public health relevance addressed?

Reviewer #2: The conclusions are backed by the data presented in the results.

Reviewer #3: (No Response)

**Editorial and Data Presentation Modifications?**

Reviewer #2: This is an interesting and challenging study that assessed the impact of oxidative environment on parasite burden in cardiomyoblasts and the effects of the Nrf2-inducer COPP on heart function in BALB/c mice infected with either DTU-II Y or DTU-I Colombian T. cruzi strains. Treatment with antioxidants CoPP, apocynin, resveratrol, and tempol reduced parasite burden in cardiomyoblasts H9C2 for both DTUI- and II-strains, while H2O2 increased it. CoPP treatment improved electrical heart function when administered during acute stage of Y-strain infection, coinciding with an overall trend towards increased survival and reduced heart parasite burden. These beneficial effects surpassed those of trypanocidal benznidazole, implying that CoPP directly affects heart physiology. CoPP treatment had beneficial impact on heart systolic function when performed during acute and evaluated during chronic stage. No impact of CoPP on heart parasite burden, electrical, or mechanical function was observed during the chronic stage of Colombian- strain infection, despite previous demonstrations of improvement with other antioxidants. Treatment with CoPP also did not improve heart function of mice chronically infected with Y-strain.These findings indicate that amastigote growth is responsive to changes in oxidative environment within heart cells regardless of the DTU source, but CoPP influence on heart parasite burden in vivo and heart function is mostly confined to the acute phase. This is very novel data.However the main limitation is that only 2 DTU's were assessed. This has been presented in the discussion and appropriately addressed.

Reviewer #3: All issues were addressed.

**Summary and General Comments**

Reviewer #2: This is a very interesting study. Of significance the finding that CoPP may improve or delay heart alterations without affecting parasite burden is very important as it dispels the belief that t. cruzi burdennplays a role in the progression of chagas cardiomyopathy. This is a very important and relevant finding

Reviewer #3: All issues were addressed.

PLOS authors have the option to publish the peer review history of their article (what does this mean?). If published, this will include your full peer review and any attached files.

Reviewer #2: **Yes: **Carlos A. Morillo

Reviewer #3: **Yes: **João Alfredo de Moraes

---

## [Editor Report · Acceptance letter]

31 Oct 2024

Dear Dr Paiva,

We are delighted to inform you that your manuscript, " Heart Function Enhancement with Nrf2-Activating Antioxidant in Acute Y-Strain Chagas Disease, Not in Chronic Colombian or Y-Strain  ," has been formally accepted for publication in PLOS Neglected Tropical Diseases.

Best regards,

Shaden Kamhawi

co-Editor-in-Chief

Paul Brindley

co-Editor-in-Chief
